# Characterization of Physicochemical Properties of Oil-in-Water Emulsions Stabilized by *Tremella fuciformis* Polysaccharides

**DOI:** 10.3390/foods11193020

**Published:** 2022-09-29

**Authors:** Furong Hou, Shuhui Yang, Xiaobin Ma, Zhiqing Gong, Yansheng Wang, Wenliang Wang

**Affiliations:** 1Key Laboratory of Novel Food Resources Processing, Key Laboratory of Agro-Products Processing Technology of Shandong Province, Ministry of Agriculture and Rural Affairs, Institute of Agro‐Food Science and Technology, Shandong Academy of Agricultural Sciences, Jinan 250100, China; 2College of Food Science and Engineering, Shandong Agricultural University, Taian 271018, China; 3Teagasc Food Research Centre, Moorepark, Fermoy, Co., P61 C996 Cork, Ireland

**Keywords:** *Tremella fuciformis* polysaccharides, emulsifying properties, rheological properties, stability

## Abstract

In this paper, emulsions stabilized by *Tremella fuciformis* polysaccharides (TFP) were prepared and the physiochemical properties were assessed. Results showed that the TFP emulsions illustrated the highest emulsifying activity (EAI) and emulsifying stability (ESI) when the concentration of TFP and oil were 0.8% and 10% (wt%). The higher pH value was in favor of the emulsifying properties, while the addition of NaCl impaired the stability, and the greater the concentration, the lower the EAI and ESI. Besides, the emulsifying and rheological properties and stability analysis were evaluated in comparison with gum arabic, pectin, and carboxymethyl cellulose emulsions. It was discovered that TFP illustrated better storage and freeze-thaw stability, which was proved by the result of zeta-potential and particle size. The rheological measurement revealed that all the emulsions behaved as pseudoplastic fluids, while TFP displayed a higher viscosity. Meanwhile, TFP emulsions tended to form a more stable network structure according to the analysis of the parameters obtained from the Herschel–Bulkley model. FTIR spectra suggested that the O-H bond could be destructed without the formation of new covalent bonds during the emulsion preparation. Therefore, this study would be of great importance for the research of emulsions stabilized by TFP as a natural food emulsifier.

## 1. Introduction

*Tremella fuciformis*, one of the colloidal fungi with high edible and medicinal nutrition, is extremely popular and widely cultivated in China on account of the abundant and excellent active ingredients, such as polysaccharides, proteins, multivitamins, and minerals [1]. For now, *Tremella fuciformis* polysaccharides (TFP) are attracting a growing concern not only because of the high content (about 60–70% of the total dry weight) [2], but also due to their brilliant physiochemical and biological properties [3,4]. Due to the different raw materials of *Tremella*, including fresh *Tremella*, dried *Tremella* fruit body, *Tremella* powder, etc., there are many extraction methods of TFP, such as hot water extraction, alkali extraction, enzymatic extraction, and physically-assisted extraction. Then the crude polysaccharide would undergo a series of the purification process for the sake of each component of the polysaccharide, including precipitation deproteinization, decolorization, desalting, separation, and purification [5]. It is generally discovered that TFP mainly includes acidic heteropolysaccharides (70–75%), where the main chain is composed of mannan linked by α-(1→3) glycosidic bonds, and glucuronic acid and xylose constitute its branched chain [6]. Furthermore, due to the polyhydroxy structure in the molecular chain structure combined with the spatial network structure coming from the molecular side chains interaction endow TFP with a remarkable thickening and emulsifying ability [7].

In recent years, TFP are increasingly causing people’s attention as a good source of natural emulsifier in oil-in-water emulsions. Basically, the mechanism of the emulsifying properties can be explained from two aspects: first, the polyhydroxy structure makes it efficient in water-holding capacity, coupled with the ability to form a hydrophilic colloid film with the surface of oil droplets through van der Waals force, it tends to prevent aggregation and increase the emulsifying ability and stability of the system [8]; second, since the TFP aqueous solution is characterized by high viscosity, it can weaken the active aggregation between oil droplets caused by Brownian motion [5]. Therefore, TFP are gifted with great potential in applying in the dairy and beverage industry as a natural food emulsifier, not only demonstrating effective emulsion ability, but also improving eminent tastes and textures [9,10].

Up to now, the studies about TFP primarily concentrate on the extraction and biological activities, the physicochemical properties are still poorly understood. It is noted that the characteristics of polysaccharides themselves and environmental changes have a crucial influence on emulsion properties, consequently bringing a variety of influences in different degrees on the quality of the products during food processing and storage. Hence, this work aims to evaluate the physicochemical properties, including emulsification properties (emulsifying activity and stability, particle size, zeta-potential), stability properties (storage and freeze-thaw stability) and rheological properties (steady-shear and dynamic viscoelastic behavior). In addition, the comparison of TFP emulsions with other commonly used polysaccharide emulsifiers (gum arabic (GA), pectin (Pec), carboxymethyl cellulose (CMC)) of these properties is further investigated. This study will provide a theoretical basis for the probable application of TFP as a natural and newfangled food emulsifier in the food industry.

## 2. Materials and Methods

### 2.1. Materials

TFP, GA, Pec and CMC were purchased from Aladdin Biochemical Technology Co., Ltd. (Shanghai, China). Palm oil was purchased from a local supermarket. All the reagents were of analytical grade.

### 2.2. Preparation of TFP Emulsions

The TFP powder was dissolved in 0.2 M phosphate buffer (PBS, pH 7.0) to obtain TFP solutions. For the preparation of emulsion, palm oil was first added into TFP solutions, and then the mixture was homogenized by a homogenizer (XHF-D Kinematica, Ningbo Xinzhi Biotechnology Co., Ltd., Ningbo, China) at 10,000 rpm for 1 min. The TFP emulsions were obtained with different ratios of palm oil (2.5, 5.0, 7.5, 10.0, 12.5 and 15.0 wt%) and TFP solutions (0.2, 0.4, 0.6, 0.8, 1.0 and 1.2 wt%).

### 2.3. Emulsifying Properties Analysis

Emulsifying properties of TFP emulsions were measured by the turbidity method as described by Olawuyi et al. [11] with a little modification. The freshly-made emulsion (100 μL) pipetted from bottom of the centrifuge tube at 0 and 10 min was added to 0.1 % sodium dodecylsulfate (SDS) solution (10 mL). Then the absorbance of the mixture was immediately measured at 500 nm by using a UV spectrophotometer (UV-6100, Shanghai Yuanyan Instrument Co., Ltd., Shanghai, China), taking 0.1 % SDS solution as a blank. The emulsifying activity index (EAI) and emulsifying stability index (ESI) were calculated as follows:(1)$$\mathrm{EAI}(m^{2}/g)=\frac{4.606\;\times \;A_{500}}{C\;\times \;\varphi \;\times \;1000}\;\times \;\mathrm{DF}$$
(2)$$\mathrm{ESI}\left(\%\right)=\frac{A_{10}}{A_{0}}\;\times \;100\%$$
where the dilution factor (DF) is 100, *C* is the concentration of TFP (g/mL), and *φ* is the oil volumetric fraction. *A*_10_ and *A*_0_ are the absorbances at 0 min and 10 min, respectively.

### 2.4. Effect of Salt and pH on the EAI and ESI of TFP Emulsions

To evaluate the effect of salt and pH on the EAI and ESI, a series of emulsions with different pH (3, 5, 7, 9, 11) and NaCl concentrations (0.2, 0.4, 0.6, 0.8, 1.0, 1.2 wt%) were prepared. pH was adjusted to different values by using HCl (3 M) or NaOH (6 M). 

### 2.5. Zeta-Potential Measurements of Different Emulsions

TFP, GA, Pec and CMC were dissolved in 0.2 M PBS (pH 7.0) solution with a final concentration of 0.8% (wt%). Then palm oil (10 wt%) was added to the above solutions to prepare the different polysaccharide emulsions with the homogenizer (10,000 rpm, 1 min). The zeta-potential of prepared emulsions was obtained using a ZS90 Nano Zetasizer (Malvern Instruments, Ltd., Malvern, UK) with dilution by phosphate buffer (0.2 M, pH 7.0). All measurements were determined in triplicate at room temperature.

### 2.6. Optical Microscopy of Different Emulsions

The general appearances of different emulsion droplets were observed using a digital optical microscope (Ci-L, Co., Tokyo, Japan) equipped with a digital camera (D7500, Nikon Co., Tokyo, Japan). The micrographs were captured at ×10 magnification.

### 2.7. Stability Analysis of Different Emulsions

#### 2.7.1. Storage Stability

The prepared emulsions were stored at room temperature for 14 days, and the EAI and particle size were tested over 0, 7, and 14 days of storage.

EAI were measured according to the methods described in Section 2.3. The particle sizes of different emulsions were measured using BT-2001 Laser Particle Size Distribution Analyzer (Dandong Baite Instrument Co., Ltd., Dandong, China). Emulsions were gradually added to the sample bath and diluted with distilled water (refractive index was 1.33) automatically. The surface-based (D_3,2_) and volume-based mean diameter (and D_4,3_) were obtained, which were expressed as follows:(3)$$D_{4,3}=\frac{\Sigma_{i}n_{i}d_{i}^{4}}{\Sigma_{i}n_{i}d_{i}^{3}}$$
(4)$$D_{3,2}=\frac{\Sigma_{i}n_{i}d_{i}^{3}}{\Sigma_{i}n_{i}d_{i}^{2}}$$
where *n_i_* is the number of droplets of diameter, *d_i_* is the droplet diameter.

#### 2.7.2. Freeze-Thaw Stability

Different emulsions were all stored at −18 °C for 24 h. After that the samples were thawed at room temperature. This freeze-thaw treatment was conducted for 3 cycles. Then the EAI and particle size were measured.

### 2.8. Rheological Properties of Different Emulsions

#### 2.8.1. Steady Shear Properties

The viscosity of the samples was measured using an HR20 Rheometer (TA Instruments Waters Corporation, New Castle, DE, USA) with a 40 mm parallel plate, the tests were carried out at a shear rate range of 0.01–600 s^−1^ at 25 °C. Effects of shear rate on shear stress of different emulsions were also measured under the same condition. Experimental data of different flow curves were collected and fitted to the Power-Law model (Equation (5)) and Herschel–Bulkley model (Equation (6)): (5)$$\gamma \eta =K\gamma^{n}$$
(6)$$\tau =\tau_{0}+K\gamma^{n}$$
where *γ* is the shear rate (s^−1^), *η* is the viscosity (Pa·s^−1^), *τ* is shear stress (Pa), τ_0_ is yield stress (Pa), K is consistency coefficient (Pa·s^n^) and n is flow behavior index.

#### 2.8.2. Dynamic Viscoelastic Behavior

First, the oscillation amplitude strain sweep ranging from 0.01% to 100% was carried out at an angular frequency of 10 rad/s to obtain the linear viscoelastic region (LVR). Further, the dynamic viscoelastic behavior of different emulsions was performed at an angular frequency range of 0.1–100 rad/s within the linear viscoelastic region (strain of 1%). The storage modulus (G′), loss modulus (G″), and loss tangent (tan δ = G″/G′) were recorded as mechanical spectra.

#### 2.8.3. Temperature Sweep

Different emulsions were performed in temperature sweep tests in the range of 25–60 °C at a scan rate of 0.5 °C/min with a strain of 1% and angular frequency of 10 rad/s.

### 2.9. Fourier Transform Infrared Spectroscopy (FTIR)

The FTIR spectrum was measured using an infrared spectrometer (Nicolet is 5, Thermo Fisher Scientific, Waltham, MA, USA). Different samples were put into the liquid cell for testing. All the samples were scanned 32 times at the wavenumber region of 400–4000 cm^−1^ at the resolution of 4 cm^−1^.

### 2.10. Statistical Analysis

All experiments were tested in triplicate and results were expressed as the mean value ± standard deviation (SD). All of the figures were processed by Origin Software Version 8.5 (Origin Lab Corp., Waltham, MA, USA), and the significant differences (*p* < 0.05) were analyzed by one-way analysis of variance (ANOVA) or mixed ANOVA followed by Duncan’s posthoc test using the SPSS 18.0 (SPSS Inc., Chicago, IL, USA).

## 3. Results and Discussions

### 3.1. Effect of TFS and Oil Concentration on EAI and ESI

The EAI indicates the fraction of oil enveloped by a continuous phase, while the ESI implies the relative stability of the emulsion at a pre-determined time [12]. Therefore, high EAI and ESI values demonstrate that more oil volume could be absorbed in the interface and the high ability of the emulsion could maintain a stable state [13]. As shown in Figure 1A, within the experimental concentration range, with the increase in TFP concentration, the EAI and ESI illustrated an upward trend, and reached the maximum of 0.98 m^2^/g and 88% at the TFP concentration of 0.8%. However, with the continuous enhancement of the TFP concentration, a slow downward trend was obtained and kept basically constant. This was attributed to the enhanced coverage on the droplet surface as concentration increased, thereby facilitating long-term stability of the emulsion by means of the formation of a thick protective layer surrounding oil droplets [14]. Nevertheless, as the concentration increased to a certain level, the emulsion was too viscous and gel-like, which was not conducive to the dispersion of oil, and it became difficult to form a stable emulsification system. 

Similarly, the EAI and ESI of TFP emulsions correspondingly boosted as the oil ratio increased (Figure 1B), illustrating the formation of an adsorption film around the droplet at high oil concentrations [15]. When the oil ratio was over 10%, there was a slow downtrend and gradually constant. Due to the constant volume of the hydrocolloid emulsifier molecules, the high content of the oil droplet could not be completely covered, resulting in the possibility of the exposure and aggregation of oil droplets [16]. Similar results were also found in rice pectin emulsions [15] and pectin-zein complex emulsions [17], the increment in the content of internal phases of the emulsion droplets was presumably to coalesce.

### 3.2. Effect of pH and Salt on EAI and ESI

The effect of pH and NaCl on EAI and ESI of TFP emulsions is given in Figure 2. In general, pH exerts an important influence on the change in the stability of emulsions due to the changes in hydrophilic-lipophilic balance caused by pH [18]. As exhibited in Figure 2A, EAI and ESI of TFS emulsions enhanced with the increase of pH from 3 to 11, indicating an increase in emulsion stability. It was discovered that TFP are acidic hetero-polysaccharides and main chains are composed of mannose [1]. Therefore, it displays a sub-acidity pH value when dissolving in water solution, as pH is reduced, more H^+^ in the water solution would be produced, leading to repulsion between the molecule chains and disability of the emulsion [14]. With the increase of the pH, the electrostatic repulsion was correspondingly decreased, and then the oil droplets had a great likelihood to aggregate at the isoelectric point, stabilizing the emulsion by the van der Waals forces [2]. Furthermore, higher pH could result in the exposure of the hydrophobic regions, conducing to a better orientation at the oil-water interface [18].

The effect of NaCl concentration on the EAI and ESI of TFP emulsions is illustrated in Figure 2B. It could be seen that with the increase of NaCl concentration, the EAI and ESI decreased, implying a change in the surface charge of the emulsion due to the addition of NaCl. At low NaCl concentrations, the TFP emulsions were stabilized by the complex particles by means of electrostatic repulsion [19], while at high NaCl concentrations, coalescence and aggregation came up because of the electrostatic reaction among the net charge of the droplets as well as the hydrophobicity of the emulsion [20].

### 3.3. Zeta-Potential and Optical Microscopy

Zeta-potential is one of the predominant indicators to investigate the electrostatic interaction between the droplet surface. It is commonly acknowledged that the higher the absolute value, the stronger the electrostatic interaction, thereby exerting a positive influence on the stability of emulsions [21]. In particular, when it is higher than 30, it can be considered that the electrostatic repulsion can make the system stable, while little repulsion is more likely to coalesce rapidly because of the less interaction between the droplet particles [22]. As displayed in Figure 3, the zeta-potential values of the different emulsions stabilized by polysaccharides ranged from −45.13 mV to −76.40 mV, indicating a strong electrostatic repulsive force. Furthermore, the absolute value of the zeta potential of TFP was significantly higher than that of other emulsions (*p* < 0.05) and then followed by Pec, ascribing to the electrostatic shielding effect of different polysaccharides [23]. The differences in zeta-potential of various emulsions stabilized by TFP, GA, Pec, and CMC could be ascribed to the diversified structure of polysaccharides, such as the molecular weight, molecular size, and conformation [24,25,26]. Furthermore, the optical micrographs of emulsions could also be confirmed by this phenomenon, verifying the results of the previous analysis of stability and particle size. Though the oil droplets of all the emulsions were spherical and the sizes of GA emulsions were even smaller (more likely to coalescence and aggregation due to the low Zeta-potential), the particle size of TFP and Pec emulsions were more evenly distributed compared with GA and CMC emulsions. 

### 3.4. Stability Analysis of Different Emulsions

#### 3.4.1. Storage Stability

Storage stability is one of the crucial indicators for evaluating the physical stability of various emulsions [27]. Figure 4A was the EAI of different emulsions placed at room temperature for 14 days, and the corresponding appearance drawings were inset. It could be seen that the EAI of all the emulsions decreased throughout the storage period. The EAI of TFP and Pec emulsions were obviously higher than GA and CMC after storage for 7 and 14 days (*p* < 0.05), indicating higher resistance to creaming and the adsorption of more oil volume at the interface [11]. While in GA and CMC emulsions, rapid phase separation and apparent destabilization occurred after 7 days of storage, which could be intuitively proved by the inset digital photo. Besides, no significant difference was obtained between storage for 7 days and 14 days of GA, CMC, and Pec emulsions (*p* > 0.05), implying a decreased stability occurred during the first week. The EAI results were consistent with the results of surface-based mean diameter (D_3,2_) (Figure 4B). As the storage period increased, the D_3,2_ of all emulsions gradually increased, while TFP emulsions were stable with the lowest D_3,2_ after 2-weeks of storage. In short, TFP emulsion demonstrated low EAI increase and excellent storage stability during the storage period.

#### 3.4.2. Freeze-Thaw Stability

The effect of freeze-thaw treatment on the apparent morphology and EAI of different emulsions is displayed in Figure 5. As can be seen from the photo in Figure 5, the delamination phenomenon of GA and CMC emulsions gradually appeared with the increase of freeze-thaw treatment times, while no obvious phase separation was observed in TFP and Pec. TFS emulsions showed the best stability after three freeze-thaw cycles, followed by Pec with less oiling-off. While oil phase was precipitated in large quantities and EAI was decreased significantly (*p* < 0.05) in CMC and GA emulsions after two freeze-thaw cycles, indicating that the damage to membranes around the droplets resulted from freeze-thaw treatments [28].

The changes in EAI might be attributed to the droplet size and homogeneity, which agreed with the results shown in Table 1. The variation of D_3,2_ of TFP, GA, Pec, and CMC between 0 and 3 freeze-thaw cycles were 24.90, 113.27, 29.56, and 74.50 μm, and the variations of D_4,3_ were 82.69, 130.33, 111.08 and 97.89 μm. As shown by the low span value of TFP, it revealed that TFP had excellent freeze-thaw stability and great potential for application in the frozen food processing industry. Furthermore, it could be concluded that different polysaccharides, freeze-thaw cycles, and their interaction effect had a very significant effect on the D_3,2_ and D_4,3_ (*p* < 0.001), while for D_3,2_, the effect from various polysaccharides was more pronounced (F = 31.80), for D_4,3_, the effect from the interaction between polysaccharides and freeze-thaw cycles was greater (F = 7.33)

### 3.5. Rheological Properties

#### 3.5.1. Steady-Shear Behavior

Viscosity is one of the most important parameters to illustrate the flow properties and will influence the appearance and consistency of emulsions. From the results of the steady-shear behavior measurements, it was found that with the increase of shear rate, the viscosity of all emulsions continuously decreased and demonstrated shear thinning behaviors (Figure 6A), implying the existence of network structure and weak interaction among the droplet-droplet particles. The shear force could not only destroy the entangled network of polysaccharides, but also disrupt the orientation of fibrils along streamlines, thereby leading to the reduction in viscosity [29]. Normally, the higher the apparent viscosity, the higher the emulsion stability. Consistent with the results of EAI of different emulsions, when compared to GA and CMC, the TFS and Pec illustrated relatively high apparent viscosity, which could be attributed to the various macromolecular compositions and polymer molecular chain orientation degrees [1,30]. However, the shear stress gradually increased especially at higher shear rates due to the disruption of O-W interfaces (Figure 6B) [16]. Like the viscosity, TFP and Pec illustrated the higher shear stress, and especially at a higher shear rate the shear stress of TFP emulsions surpassed Pec emulsions, stating that under external force less deformation and stronger interaction force of TFP was observed compared with Pec [31]. This phenomenon could be perfectly verified by the result of the viscosity, since strong shear stress could give rise to the emulsion droplets rearrangement, which not only amplified the steric hindrance effect but also intensified the intermolecular friction among polysaccharides, consequently, the viscosity was enhanced [32].

In order to evaluate the stability mechanism, the flow curve parameters were fitted to the Power-Law model (Table 2) and the Herschel–Bulkley model (Table 3) to explain the droplet-droplet interaction. In the Power-Law model, *n* is the fluid index, which characterizes the type of fluid. It could be seen that all the obtained *n* values were less than 1, indicating that all emulsions were pseudoplastic fluids. K is the consistency coefficient, which characterizes the appearance and viscosity of the fluid. The smaller the value, the weaker the viscosity-increasing ability. As shown in Table 2, Pec illustrated the highest K value, followed by TFP, CMC, and GA, which was in agreement with the viscosity results. This could have resulted from the different droplet sizes in various emulsions, where the bigger the droplet size, the less availability of droplets to interact, therefore leading to a low viscosity [33,34]. However, the R^2^ of all the emulsions in the Herschel–Bulkley model are higher (Table 3) than that in the Power-Law model, indicating a better fitting effect (R^2^ > 0.9). The existence of τ_0_ implies that an initial force is necessary to start the flowing, which has a direct relation to fluidity. Generally, a higher value of τ_0_ illustrates a higher ability to sustain its appearance and stability [35]. It is known that when the applied stress is higher than τ_0_, the emulsion begins to flow, so τ_0_ is usually used to forecast the stability of the emulsion network structure [36]. As can be seen in Table 3, TFP and Pec presented higher τ_0_ compared to GA and CMC, showing a lower fluidity with high viscosity. Besides, higher K and lower *n* values are considered important indexes in a stable emulsion system. In summary, in contrast to GA, Pec, and CMC, the rheological results exhibited that TFP had the capacity to form a more stable network structure by resisting the droplets coalescence effectively, accordingly improving the emulsion stability [23].

#### 3.5.2. Frequency Sweep

Before frequency sweep, the oscillation amplitude strain sweep experiment (0.01–100%) was carried out at an angular frequency of 10 rad/s to obtain the linear viscoelastic region (LVR). It was observed that different emulsions demonstrated a constant storage modulus (G′) and viscous modulus (G″) in the oscillatory stress range of 0.01–1% (data not shown). Then the frequency sweep was conducted at an angular frequency of 0.1 to 100 rad/s at a fixed oscillation strain of 1% (within the LVR) to study the viscous and elastic behavior changes. For Pec and GA emulsions, no crossover point (G′ = G″) was discovered and G′ was always higher than G″, manifesting gel-like behavior with more accessibility to an ordered network structure [35]. While for TFP and CMC, there was a crossover point (G′ = G″) within the tested range. It could be seen in Figure 7A that G″ of TFP and CMC emulsions were higher than G′ at low frequency, showing a viscous characteristic of a liquid. This owed to the sufficient stress relaxation time of the polymers allowed them to deform slowly, and then most of the energy would be dissipated, leaving a little energy in the emulsion [31]. However, G′ exceeded G″ at high frequency, illustrating an elastic characteristic similar to a solid. This was because the relaxation time was cut down with the enhancement of frequency, and a solid-like network structure was gradually generated, thereby preserving the energy until G′ > G″ [31]. 

Furthermore, the elastic and viscous properties of different emulsions are characterized by the ratio of G″ to G′ (tan δ) [18]. Generally, there is an elastic behavior when G′ is larger than G″ (tan δ < 1), whereas when G′ is smaller than G″, implying a viscous behavior [37]. As displayed in Figure 7B, tan δ at the crossover point of TFP and CMC emulsions were 0.92 and 0.91 (both were greater than 0.1 but lower than 1), exhibiting a weak gel behavior.

#### 3.5.3. Frequency Sweep

In order to evaluate the sensitivity of different emulsions to temperature changes, a temperature sweep was carried out. As illustrated in Figure 8, melting was found in TFP emulsion during the temperature range, demonstrating that there was a transformation from a solid state to a liquid state as temperature increased. It was also discovered that Pec emulsion had dominance over elastic behavior (G′ > G″) and formed structured gels, while CMC demonstrated a viscous behavior (G″ > G′). Though GA showed similar results to Pec (G′ > G″), the G′ and G″ values were quite low, which resulted from the low probability of collision between particles [38,39].

### 3.6. FTIR

FTIR was applied to investigate the structural change of different polysaccharides after the emulsion preparation. As demonstrated in Figure 9, there was a strong infrared absorption peak in region ranging from 3600 cm^−1^ to 3000 cm^−1^ of all the polysaccharides, indicating an O-H stretch region, while these peaks disappeared of all the emulsions, showing the destruction of the O-H bond during the emulsion preparation [40]. Furthermore, absorption peaks at around 1745 cm^−1^, 2854 cm^−1^ and 2925 cm^−1^ of all emulsions became more visible in contrast with the corresponding polysaccharides, where the peaks were assigned to the stretching vibrations of the C=O, asymmetric and symmetric stretching of -CH_3_ or -CH_2_, respectively [41]. Similar results were found by Liu et al. [42] that these peaks of emulsion gels stabilized by xanthan and sodium stearoyl lactylate were intensified, it was presumed that the potential hydrophobic interactions occurred, which was advantageous for this phenomenon on account of the molecular structure. Also, absorption peaks between 1000 cm^−1^ and 1100 cm^−1^ were attributed to the vibrations of the C-O-C or C-O-H group, all of which were found in the polysaccharides and emulsions [1]. On the whole, there were no new absorption peaks in emulsions according to FTIR spectra, showing no new covalent binding appeared during the emulsion preparation. Furthermore, it was found that FTIR spectra could be substantially modified during the formation of the emulsion owing to the hydrogen bond interactions between polysaccharides and emulsions, therefore demonstrating similar FTIR spectra of different emulsions [43,44]. For example, the addition of polysaccharides would increase the intensity of the aliphatic CH_2_ group, including the anti- or symmetric stretching (2854 cm^−1^ and 2925 cm^−1^) and rocking vibration (720 cm^−1^) [45].

## 4. Conclusions

In this study, emulsions stabilized by TFP were produced, and their physicochemical properties (emulsification, stability, and rheological properties) were systematically evaluated. It was found that the EAI and ESI of TFP emulsion were influenced by TFP and oil concentration, as well as pH value and salt concentration. When the TFP and oil concentrations were 0.8% and 10% (wt%), the maximum values of EAI and ESI of TFP emulsions were obtained. Besides, higher pH was beneficial to the emulsifying properties, while the addition of NaCl exerted a negative effect on it. In contrast with GA, Pec, and CMC emulsions, TFP illustrated the best freeze-thaw and storage stability with the highest zeta-potential (absolute value) and lowest particle size. According to the analysis of steady-shear behavior, the TFP and Pec demonstrated higher viscosity and could be effectively fitted by the Herschel–Bulkley model (R^2^ > 0.9). The results from the dynamic viscoelastic behavior revealed that TFP emulsions exhibited a weak gel behavior, and a transformation from solid state to liquid state was also discovered as temperature increased. Results of FTIR spectra implied that during the emulsion preparation the O-H bond would be damaged, while no formation of new covalent bonds were observed. Hence, the outcome of this research clearly clarifies that TFP have a great potential to be an excellent natural polysaccharide emulsifier in the food industry.

## Figures and Tables

**Figure 1 foods-11-03020-f001:**
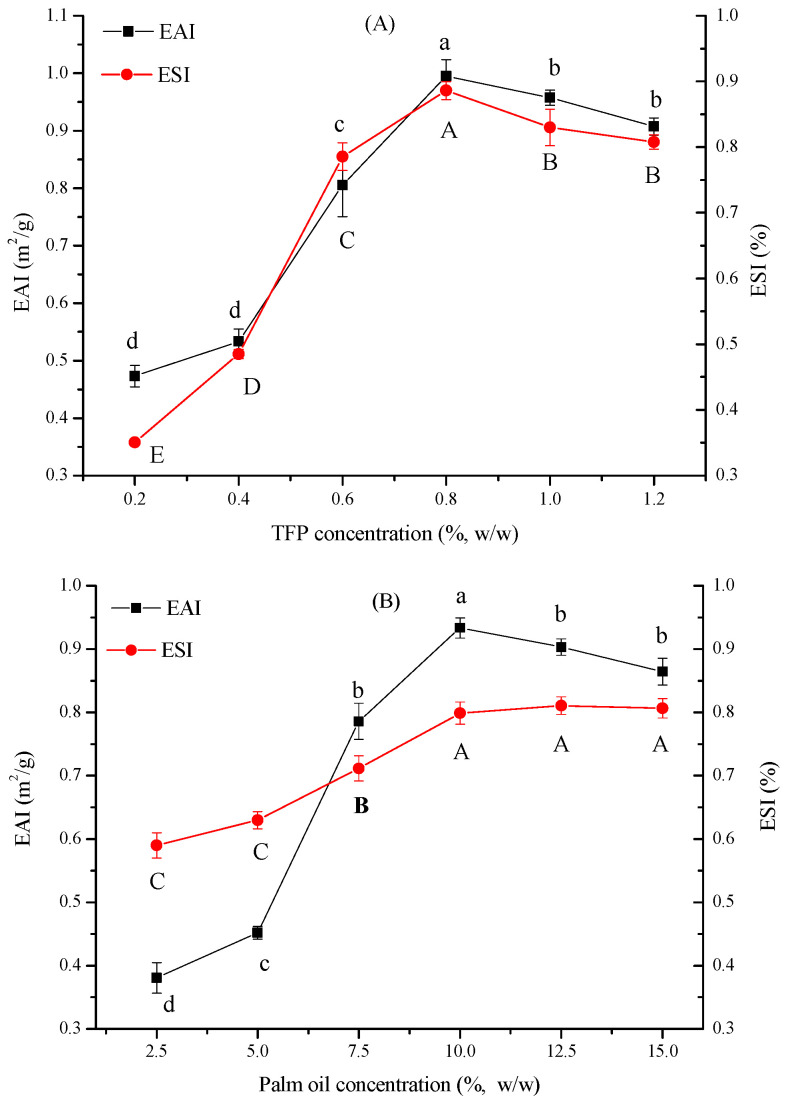
Effect of TFP (**A**) and oil (**B**) concentration on EAI and ESI. (Different lowercase and uppercase letters indicate a significant difference between EAI and ESI at *p* < 0.05).

**Figure 2 foods-11-03020-f002:**
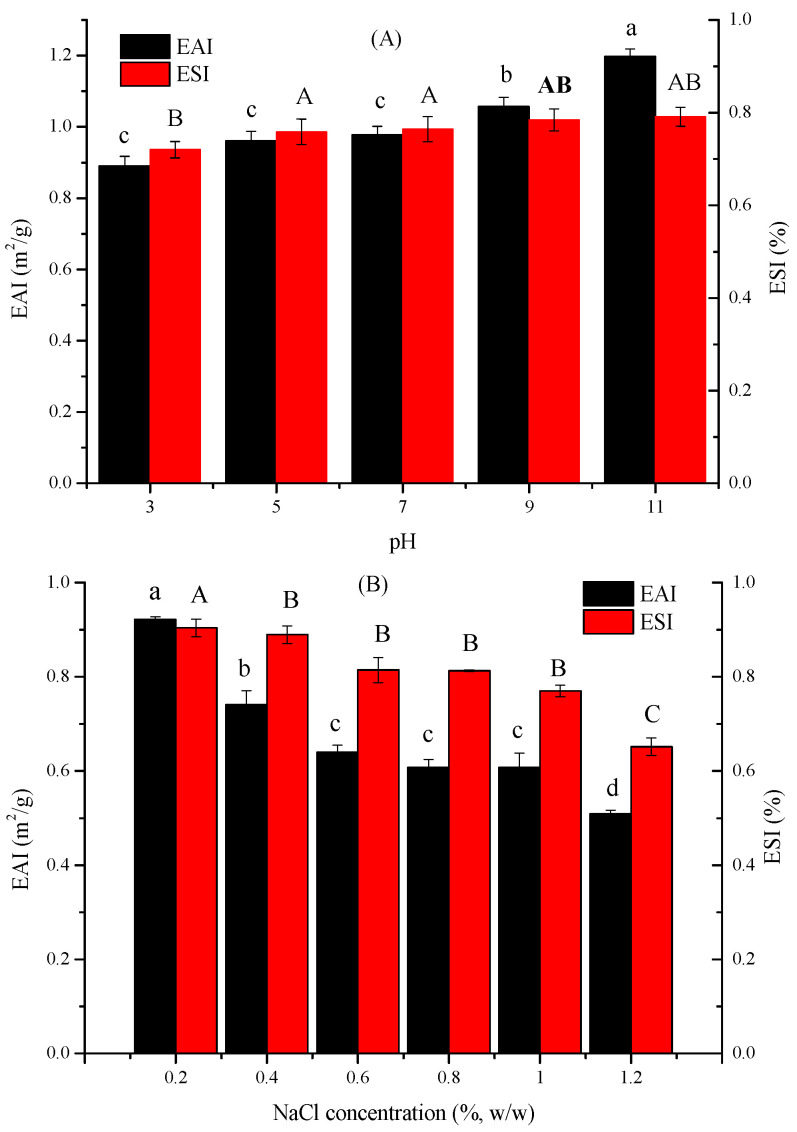
Effect of pH (**A**) and NaCl (**B**) on EAI and ESI. (Different lowercase and uppercase letters indicate a significant difference between EAI and ESI at *p* < 0.05).

**Figure 3 foods-11-03020-f003:**
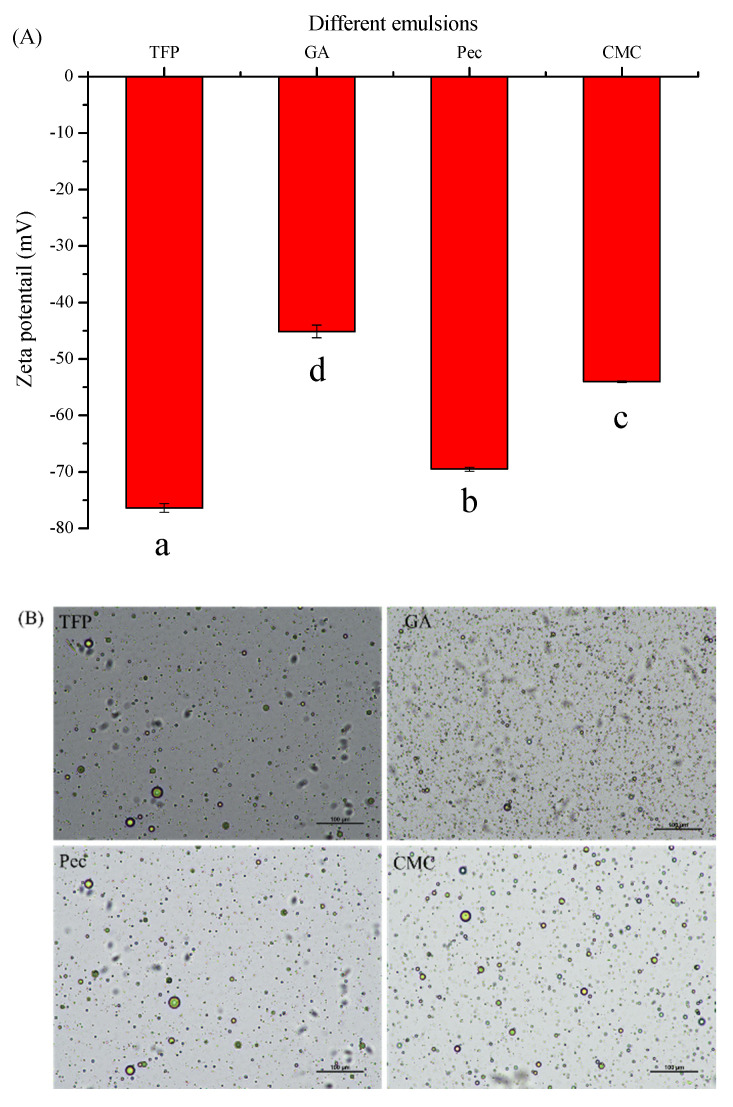
Optical microscopy (**A**) and zeta-potential (**B**) of different emulsions. (Different lowercase letters in (**A**) indicate a significant difference of zeta potential at *p* < 0.05).

**Figure 4 foods-11-03020-f004:**
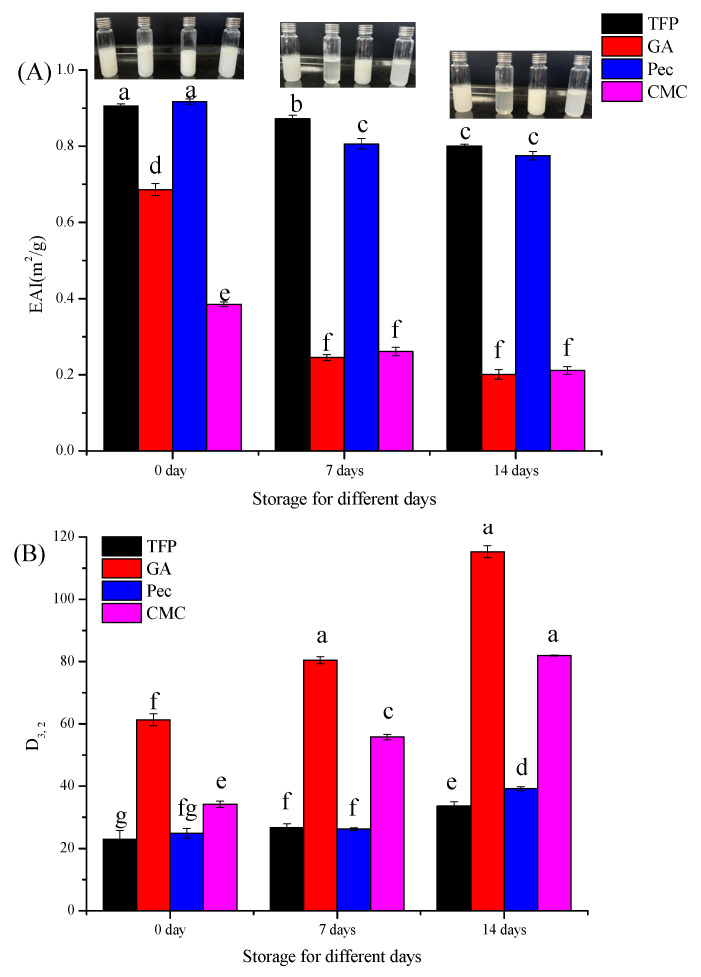
The EAI (**A**) and D_3,2_ (**B**) of different emulsions after storage for 0, 7, and 14 days at room temperature. (Different lowercase letters in indicate a significant difference of EAI (**A**) and D_3, 2_ (**B**) at *p* < 0.05).

**Figure 5 foods-11-03020-f005:**
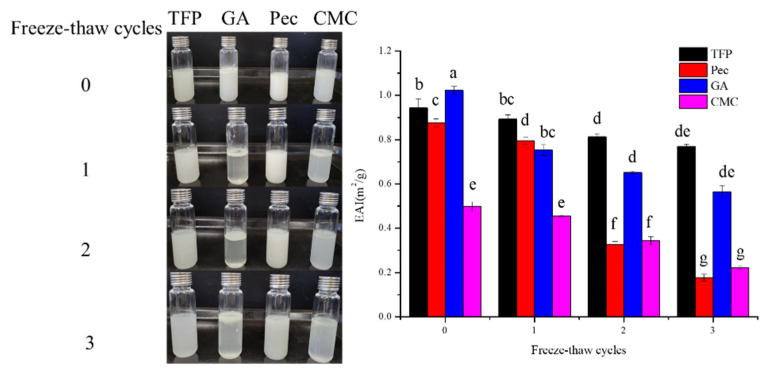
Effect of freeze-thaw treatment on the stability of different emulsions (Different letters indicate significant difference at *p* < 0.05).

**Figure 6 foods-11-03020-f006:**
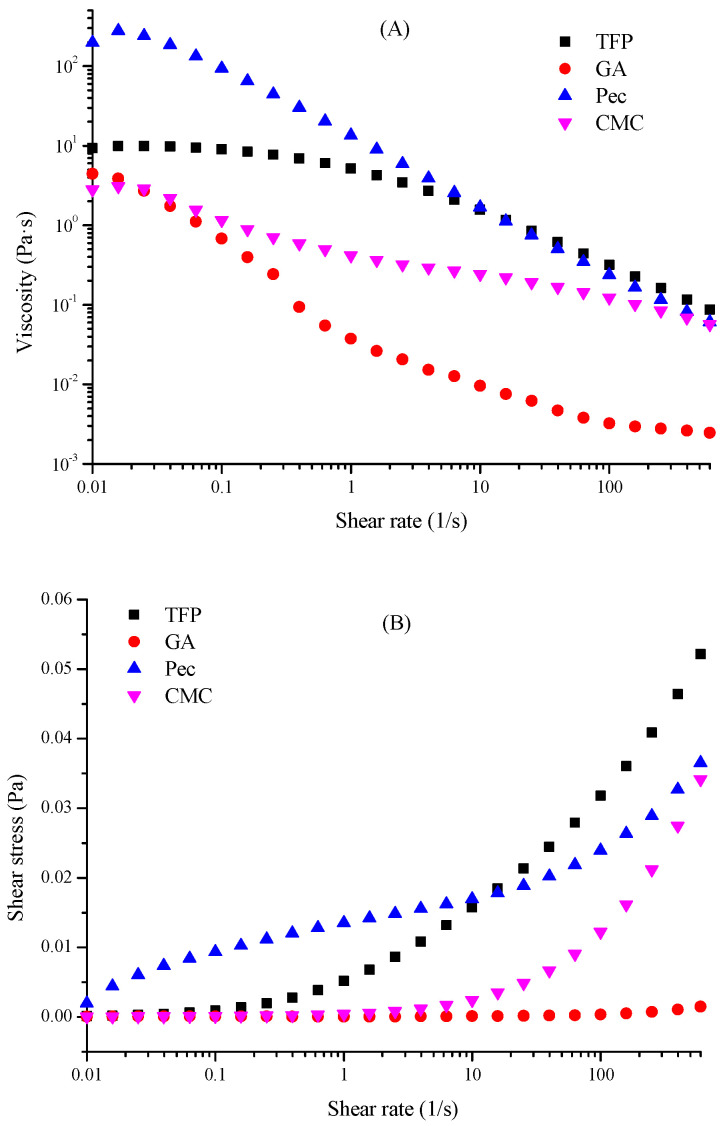
Rheological properties of different emulsions (**A**) Viscosity-shear rate relationship; (**B**) Shear stress-shear rate relationship.

**Figure 7 foods-11-03020-f007:**
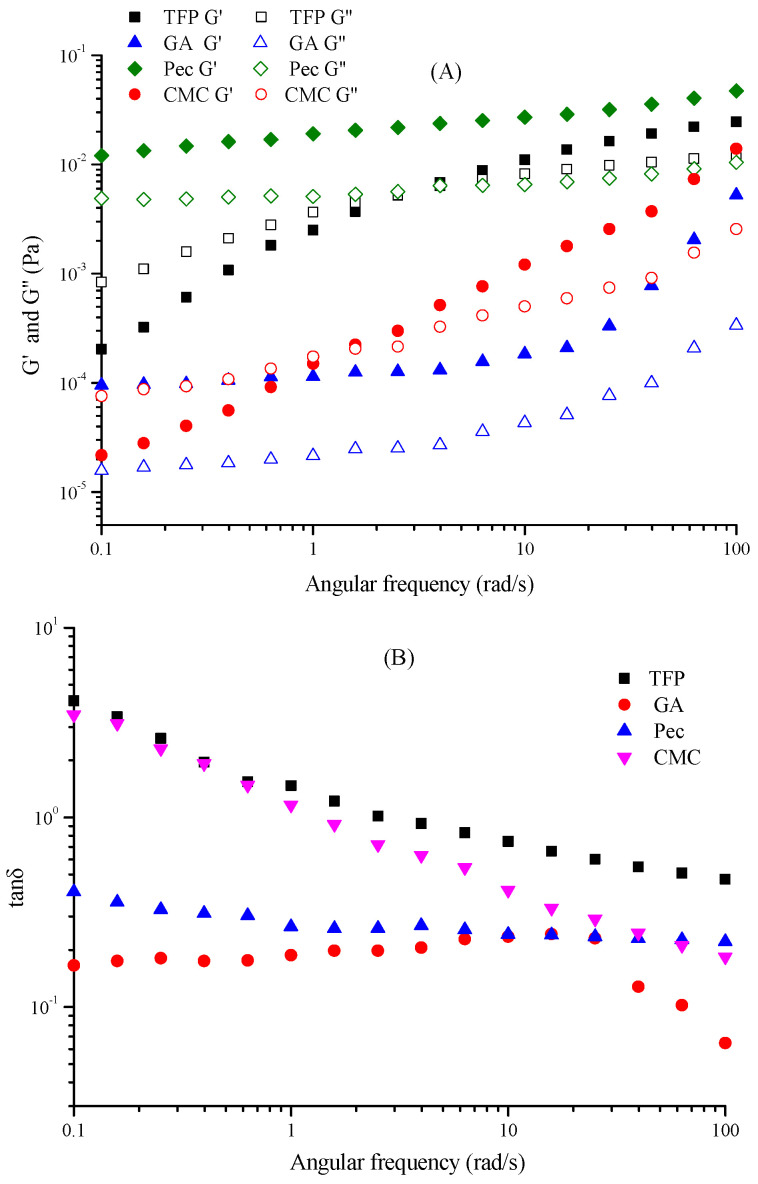
The G′ (solid symbols) and G″ (open symbols) (**A**) and tan δ (**B**) of different emulsions with the relationship of angular frequency.

**Figure 8 foods-11-03020-f008:**
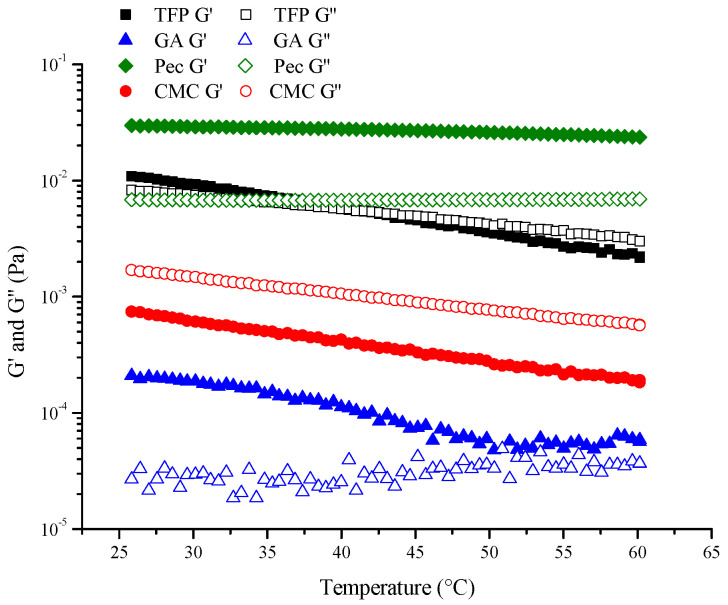
Temperature sweeps (heating) for different emulsions.

**Figure 9 foods-11-03020-f009:**
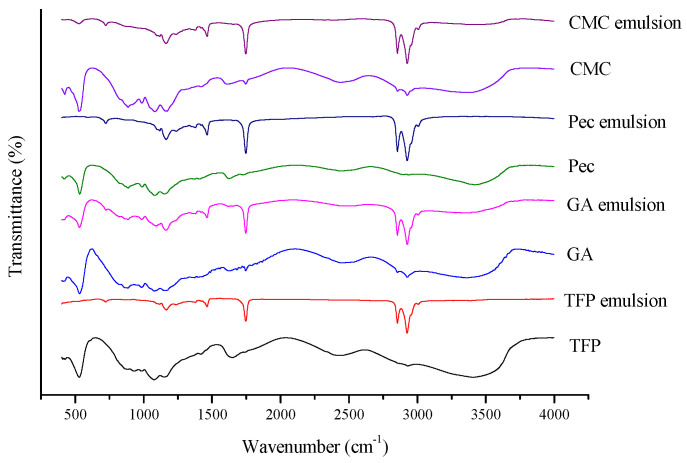
FTIR of TFP, GA, Pec, and CMC and its corresponding emulsions.

**Table 1 foods-11-03020-t001:** Effect of freeze-thaw treatment on the particle size of different emulsions.

Samples	D_3,2_ (μm)				F-Value	*p*	D_4,3_ (μm)				F-Value	*p*
	Freeze-Thaw Cycles						Freeze-Thaw Cycles					
	0	1	2	3			0	1	2	3		
TFP	11.75 ± 1.13 ^a^	23.95 ± 2.04 ^b^	27.15 ± 0.7 ^c^	36.65 ± 1.66 ^c^			32.90 ± 7.17 ^C^	99.22 ± 3.53 ^B^	104.99 ± 0.40 ^A^	115.59 ± 0.73 ^B^		
GA	3.67 ± 0.82 ^c^	60.81 ± 0.40 ^a^	80.11 ± 5.56 ^a^	116.94 ± 1.46 ^a^			22.08 ± 0.90 ^D^	53.50 ± 3.25 ^D^	118.79 ± 0.83 ^S^	152.41 ± 6.53 ^A^		
Pec	9.38 ± 0.17 ^b^	26.59 ± 1.74 ^b^	27.12 ± 0.72 ^c^	38.94 ± 0.19 ^c^			37.36 ± 0.32 ^B^	75.32 ± 0.47 ^C^	87.63 ± 2.28 ^B^	148.44 ± 3.53 ^A^		
CMC	8.19 ± 1.62 ^b^	33.09 ± 0.64 ^b^	35.12 ± 0.71 ^b^	82.69 ± 1.24 ^b^			56.20 ± 4.50 ^A^	115.59 ± 0.73 ^A^	112.18 ± 1.24 ^A^	154.09 ± 3.53 ^A^		
X					31.80	<0.001					2.32	<0.001
Y					4.29	<0.001					4.27	<0.001
X*Y					5.55	<0.001					7.33	<0.001

Note: Different uppercase and lowercase letters indicate significant differences of D_3,2_ and D_4,3_ column-wise at *p* < 0.05. Different emulsions and free-thaw cycles are named factor X and factor Y.

**Table 2 foods-11-03020-t002:** Power-Law model fitting parameters for rheological curves of different emulsions.

Samples	K/Pa·s^n^	*n*	R^2^
TFP	4.3639 ± 0.3361	0.7763 ± 0.0229	0.8814
GA	0.1467 ± 0.0238	0.2420 ± 0.0387	0.9814
Pec	32.8583 ± 6.9854	0.5342 ± 0.0546	0.8941
CMC	0.5096 ± 0.0548	0.5908 ± 0.0283	0.9536

**Table 3 foods-11-03020-t003:** Herschel–Bulkley model fitting parameters for rheological curves of different emulsions.

Samples	τ_0_/Pa	K/Pa·s^n^	*n*	R^2^
TFP	(4.8200 ± 0.6160) × 10^−3^	(11.0400 ± 0.6730) × 10^−3^	0.2577 ± 0.0083	0.9975
GA	(0.0545 ± 0.0035) × 10^−3^	(0.0043 ± 0.0004) × 10^−3^	0.9083 ± 0.0154	0.9985
Pec	(0.4190 ± 2.7000) × 10^−3^	(12.1300 ± 2.8500) × 10^−3^	0.1606 ± 0.0273	0.9677
CMC	(0.2830 ± 0.0998) × 10^−3^	(0.8110 ± 0.0459) × 10^−3^	0.5887 ± 0.0092	0.9988

## Data Availability

The date are available from the corresponding author.

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
