# Peer review of "Characterization of Physicochemical Properties of Oil-in-Water Emulsions Stabilized by Tremella fuciformis Polysaccharides"

_foods, 2022, doi:10.3390/foods11193020_

Round 1

Reviewer 1 Report

This study characterized the physicochemical properties of oil-in-water emulsion stabilized by Tremella fuciformis polysaccharides. The manuscript is well written and the results are clear and interesting.

Comments:

For all tables and figures, please use the letter "a" for the largest value.

L 35; …due to “their” brilliant physiochemical…

L 105; Please add more details about the digital camera (model, country, etc.).

L 146; Section 2.9. Please add more details about the FTIR experiment, such as resolution, spectra range, etc.

L 207; Figure 2. Please add the significant letters, and check the label of Fig. 2-B (F1 and D)

L 222-224; What did the authors exactly notice from the Optical Microscopy? Please explain more.

L 233-235; The EAI was not significantly different between TFP and Pec on days 0 and 14.

Author Response

Dear reviewer,

On behalf of my co-authors, we thank you very much for giving us an opportunity to revise our manuscript. We appreciate editor and reviewers for their positive and constructive comments and suggestions on our manuscript entitled " Characterization of Physicochemical Properties of Oil-In-Water Emulsions Stabilized by Tremella Fuciformis Polysaccharides", " (Manuscript ID: foods-1913887). According to the comments raised by the you, we gave the corresponding responses and made revisions using the “Track Changes” function in the manuscript. Please see the attachment.

We would like to express our great appreciation to you for comments to our paper. Looking forward to hearing from you!

Thank you and best regards.

Yours sincerely,

Furong Hou

Address: Institute of Agro-Food Sciences and Technology, Shandong Academy of Agricultural Sciences, 202 Gongye North Road, Jinan, Shandong 250100, China.

Reviewer 2 Report

In this manuscript, authors developed Tremella fuciformis polysaccharides (TFP) emulsions and revealed the physiochemical properties of the emulsion. Authors compared the results of TFP emulsion with GA, Pec, CMC emulsion. The manuscript was written well with detailed analysis. Following are my queries and suggestion:

Authors can briefly explain how this Tremella fuciformis polysaccharides are manufactured from Tremella fuciformis polysaccharides in the introduction. It will be interesting for the readers.

Line 68: Authors can give the full form for GA, Pec and CMC when it appears first in the manuscript.  

Line 75: Seems the XHF-D Kinematica homogenizer is a high speed homogenizer. Does just 1 min high speed stirring at 10000 rpm was sufficient to make the stable emulsion?

What is the difference between the emulsion prepared in section 2.2 and section 2.5.

Section 2.10.: Authors can detail the statistical procedure. For instance authors did not mentioned the details of the ANOVA and post hoc test for statistical significance.

Figure 2a: Inside figure 2a, FI and D was mentioned. Author need to correct that.

Section 3.4.1, 3.4.2: From Figure 4, 5 and Table 1, looks author conducted Anova for the entire dataset i.e., between the four emulsion groups and three storage time. What is the author’s finding with this experimental design. I believe, author may need to do mixed ANOVA model between the impact of storage period and impact of stabilizers.

Section 3.6: Authors need to explain how the FTIR peaks of CMC emulsion, Pec emulsion, GA emulsion and TFP emulsion were quite similar. 

Author Response

Dear reviewer,

On behalf of my co-authors, we thank you very much for giving us an opportunity to revise our manuscript. We appreciate editor and reviewers for their positive and constructive comments and suggestions on our manuscript entitled " Characterization of Physicochemical Properties of Oil-In-Water Emulsions Stabilized by Tremella Fuciformis Polysaccharides", " (Manuscript ID: foods-1913887). According to the comments raised by you, we gave the corresponding responses and made revisions using the “Track Changes” function in the manuscript. Please see the attachment.

We would like to express our great appreciation to you for comments to our paper. Looking forward to hearing from you!

Thank you and best regards.

Yours sincerely,

Furong Hou

Address: Institute of Agro-Food Sciences and Technology, Shandong Academy of Agricultural Sciences, 202 Gongye North Road, Jinan, Shandong 250100, China.

Round 2

Reviewer 2 Report

Authors modified the manuscript, and addressed all the queries.